# End-To-End Memory Networks

**Sainbayar Sukhbaatar**
Dept. of Computer Science
Courant Institute, New York University
sainbar@cs.nyu.edu

**Arthur Szlam    Jason Weston    Rob Fergus**
Facebook AI Research
New York
{aszlam,jase,robfergus}@fb.com

## Abstract

We introduce a neural network with a recurrent attention model over a possibly large external memory. The architecture is a form of Memory Network [23] but unlike the model in that work, it is trained end-to-end, and hence requires significantly less supervision during training, making it more generally applicable in realistic settings. It can also be seen as an extension of RNNsearch [2] to the case where multiple computational steps (hops) are performed per output symbol. The flexibility of the model allows us to apply it to tasks as diverse as (synthetic) question answering [22] and to language modeling. For the former our approach is competitive with Memory Networks, but with less supervision. For the latter, on the Penn TreeBank and Text8 datasets our approach demonstrates comparable performance to RNNs and LSTMs. In both cases we show that the key concept of multiple computational hops yields improved results.

## 1   Introduction

Two grand challenges in artificial intelligence research have been to build models that can make multiple computational steps in the service of answering a question or completing a task, and models that can describe long term dependencies in sequential data.

Recently there has been a resurgence in models of computation using explicit storage and a notion of attention [23, 8, 2]; manipulating such a storage offers an approach to both of these challenges. In [23, 8, 2], the storage is endowed with a continuous representation; reads from and writes to the storage, as well as other processing steps, are modeled by the actions of neural networks.

In this work, we present a novel recurrent neural network (RNN) architecture where the recurrence reads from a possibly large external memory multiple times before outputting a symbol. Our model can be considered a continuous form of the Memory Network implemented in [23]. The model in that work was not easy to train via backpropagation, and required supervision at each layer of the network. The continuity of the model we present here means that it can be trained end-to-end from input-output pairs, and so is applicable to more tasks, i.e. tasks where such supervision is not available, such as in language modeling or realistically supervised question answering tasks. Our model can also be seen as a version of RNNsearch [2] with multiple computational steps (which we term "hops") per output symbol. We will show experimentally that the multiple hops over the long-term memory are crucial to good performance of our model on these tasks, and that training the memory representation can be integrated in a scalable manner into our end-to-end neural network model.

## 2   Approach

Our model takes a discrete set of inputs $x_1, ..., x_n$ that are to be stored in the memory, a query $q$, and outputs an answer $a$. Each of the $x_i$, $q$, and $a$ contains symbols coming from a dictionary with $V$ words. The model writes all $x$ to the memory up to a fixed buffer size, and then finds a continuous representation for the $x$ and $q$. The continuous representation is then processed via multiple hops to output $a$. This allows backpropagation of the error signal through multiple memory accesses back to the input during training.

## 2.1 Single Layer

We start by describing our model in the single layer case, which implements a single memory hop operation. We then show it can be stacked to give multiple hops in memory.

**Input memory representation:** Suppose we are given an input set $x_1, .., x_i$ to be stored in memory. The entire set of $\{x_i\}$ are converted into memory vectors $\{m_i\}$ of dimension $d$ computed by embedding each $x_i$ in a continuous space, in the simplest case, using an embedding matrix $A$ (of size $d \times V$). The query $q$ is also embedded (again, in the simplest case via another embedding matrix $B$ with the same dimensions as $A$) to obtain an internal state $u$. In the embedding space, we compute the match between $u$ and each memory $m_i$ by taking the inner product followed by a softmax:

$$p_i = \text{Softmax}(u^T m_i). \tag{1}$$

where $\text{Softmax}(z_i) = e^{z_i} / \sum_j e^{z_j}$. Defined in this way $p$ is a probability vector over the inputs.

**Output memory representation:** Each $x_i$ has a corresponding output vector $c_i$ (given in the simplest case by another embedding matrix $C$). The response vector from the memory $o$ is then a sum over the transformed inputs $c_i$, weighted by the probability vector from the input:

$$o = \sum_i p_i c_i. \tag{2}$$

Because the function from input to output is smooth, we can easily compute gradients and back-propagate through it. Other recently proposed forms of memory or attention take this approach, notably Bahdanau *et al.* [2] and Graves *et al.* [8], see also [9].

**Generating the final prediction:** In the single layer case, the sum of the output vector $o$ and the input embedding $u$ is then passed through a final weight matrix $W$ (of size $V \times d$) and a softmax to produce the predicted label:

$$\hat{a} = \text{Softmax}(W(o + u)) \tag{3}$$

The overall model is shown in Fig. 1(a). During training, all three embedding matrices $A$, $B$ and $C$, as well as $W$ are jointly learned by minimizing a standard cross-entropy loss between $\hat{a}$ and the true label $a$. Training is performed using stochastic gradient descent (see Section 4.2 for more details).

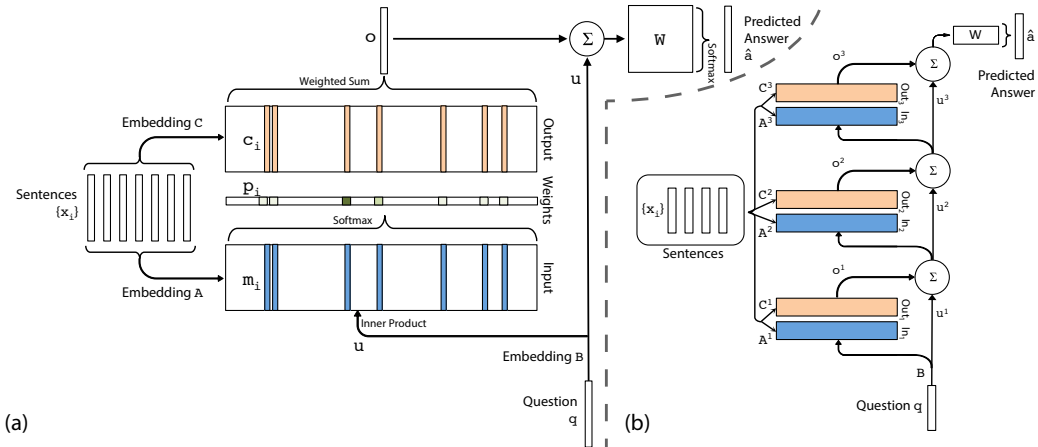

Figure 1: (a): A single layer version of our model. (b): A three layer version of our model. In practice, we can constrain several of the embedding matrices to be the same (see Section 2.2).

## 2.2 Multiple Layers

We now extend our model to handle $K$ hop operations. The memory layers are stacked in the following way:

- The input to layers above the first is the sum of the output $o^k$ and the input $u^k$ from layer $k$ (different ways to combine $o^k$ and $u^k$ are proposed later):

$$u^{k+1} = u^k + o^k. \tag{4}$$

- Each layer has its own embedding matrices $A^k, C^k$, used to embed the inputs $\{x_i\}$. However, as discussed below, they are constrained to ease training and reduce the number of parameters.
- At the top of the network, the input to $W$ also combines the input and the output of the top memory layer: $\hat{a} = \text{Softmax}(Wu^{K+1}) = \text{Softmax}(W(o^K + u^K))$.

We explore two types of weight tying within the model:

1. **Adjacent:** the output embedding for one layer is the input embedding for the one above, i.e. $A^{k+1} = C^k$. We also constrain (a) the answer prediction matrix to be the same as the final output embedding, i.e $W^T = C^K$, and (b) the question embedding to match the input embedding of the first layer, i.e. $B = A^1$.
2. **Layer-wise (RNN-like):** the input and output embeddings are the same across different layers, i.e. $A^1 = A^2 = ... = A^K$ and $C^1 = C^2 = ... = C^K$. We have found it useful to add a linear mapping $H$ to the update of $u$ between hops; that is, $u^{k+1} = Hu^k + o^k$. This mapping is learnt along with the rest of the parameters and used throughout our experiments for layer-wise weight tying.

A three-layer version of our memory model is shown in Fig. 1(b). Overall, it is similar to the Memory Network model in [23], except that the hard max operations within each layer have been replaced with a continuous weighting from the softmax.

Note that if we use the layer-wise weight tying scheme, our model can be cast as a traditional RNN where we divide the outputs of the RNN into *internal* and *external* outputs. Emitting an internal output corresponds to considering a memory, and emitting an external output corresponds to predicting a label. From the RNN point of view, $u$ in Fig. 1(b) and Eqn. 4 is a hidden state, and the model generates an internal output $p$ (attention weights in Fig. 1(a)) using $A$. The model then ingests $p$ using $C$, updates the hidden state, and so on[1]. Here, unlike a standard RNN, we explicitly condition on the outputs stored in memory during the $K$ hops, and we keep these outputs soft, rather than sampling them. Thus our model makes several computational steps before producing an output meant to be seen by the "outside world".

## 3 Related Work

A number of recent efforts have explored ways to capture long-term structure within sequences using RNNs or LSTM-based models [4, 7, 12, 15, 10, 1]. The memory in these models is the state of the network, which is latent and inherently unstable over long timescales. The LSTM-based models address this through local memory cells which lock in the network state from the past. In practice, the performance gains over carefully trained RNNs are modest (see Mikolov *et al.* [15]). Our model differs from these in that it uses a global memory, with shared read and write functions. However, with layer-wise weight tying our model can be viewed as a form of RNN which only produces an output after a fixed number of time steps (corresponding to the number of hops), with the intermediary steps involving memory input/output operations that update the internal state.

Some of the very early work on neural networks by Steinbuch and Piske[19] and Taylor [21] considered a memory that performed nearest-neighbor operations on stored input vectors and then fit parametric models to the retrieved sets. This has similarities to a single layer version of our model.

Subsequent work in the 1990's explored other types of memory [18, 5, 16]. For example, Das *et al.* [5] and Mozer *et al.* [16] introduced an explicit stack with push and pop operations which has been revisited recently by [11] in the context of an RNN model.

Closely related to our model is the Neural Turing Machine of Graves *et al.* [8], which also uses a continuous memory representation. The NTM memory uses both content and address-based access, unlike ours which only explicitly allows the former, although the temporal features that we will introduce in Section 4.1 allow a kind of address-based access. However, in part because we always write each memory sequentially, our model is somewhat simpler, not requiring operations like sharpening. Furthermore, we apply our memory model to textual reasoning tasks, which qualitatively differ from the more abstract operations of sorting and recall tackled by the NTM.

Our model is also related to Bahdanau *et al.* [2]. In that work, a bidirectional RNN based encoder and gated RNN based decoder were used for machine translation. The decoder uses an attention model that finds which hidden states from the encoding are most useful for outputting the next translated word; the attention model uses a small neural network that takes as input a concatenation of the current hidden state of the decoder and each of the encoders hidden states. A similar attention model is also used in Xu *et al.* [24] for generating image captions. Our "memory" is analogous to their attention mechanism, although [2] is only over a single sentence rather than many, as in our case. Furthermore, our model makes several hops on the memory before making an output; we will see below that this is important for good performance. There are also differences in the architecture of the small network used to score the memories compared to our scoring approach; we use a simple linear layer, whereas they use a more sophisticated gated architecture.

We will apply our model to language modeling, an extensively studied task. Goodman [6] showed simple but effective approaches which combine $n$-grams with a cache. Bengio *et al.* [3] ignited interest in using neural network based models for the task, with RNNs [14] and LSTMs [10, 20] showing clear performance gains over traditional methods. Indeed, the current state-of-the-art is held by variants of these models, for example very large LSTMs with Dropout [25] or RNNs with diagonal constraints on the weight matrix [15]. With appropriate weight tying, our model can be regarded as a modified form of RNN, where the recurrence is indexed by memory lookups to the word sequence rather than indexed by the sequence itself.

## 4  Synthetic Question and Answering Experiments

We perform experiments on the synthetic QA tasks defined in [22] (using version 1.1 of the dataset). A given QA task consists of a set of statements, followed by a question whose answer is typically a single word (in a few tasks, answers are a set of words). The answer is available to the model at training time, but must be predicted at test time. There are a total of 20 different types of tasks that probe different forms of reasoning and deduction. Here are samples of three of the tasks:

```
Sam walks into the kitchen.    Brian is a lion.            Mary journeyed to the den.
Sam picks up an apple.         Julius is a lion.           Mary went back to the kitchen.
Sam walks into the bedroom.    Julius is white.            John journeyed to the bedroom.
Sam drops the apple.           Bernhard is green.          Mary discarded the milk.
Q: Where is the apple?         Q: What color is Brian?     Q: Where was the milk before the den?
A. Bedroom                     A. White                    A. Hallway
```

Note that for each question, only some subset of the statements contain information needed for the answer, and the others are essentially irrelevant distractors (e.g. the first sentence in the first example). In the Memory Networks of Weston *et al.* [22], this *supporting subset* was explicitly indicated to the model during training and the key difference between that work and this one is that this information is no longer provided. Hence, the model must deduce for itself at training and test time which sentences are relevant and which are not.

Formally, for one of the 20 QA tasks, we are given example problems, each having a set of $I$ sentences $\{x_i\}$ where $I \leq 320$; a question sentence $q$ and answer $a$. Let the $j$th word of sentence $i$ be $x_{ij}$, represented by a one-hot vector of length $V$ (where the vocabulary is of size $V = 177$, reflecting the simplistic nature of the QA language). The same representation is used for the question $q$ and answer $a$. Two versions of the data are used, one that has 1000 training problems per task and a second larger one with 10,000 per task.

### 4.1  Model Details

Unless otherwise stated, all experiments used a $K = 3$ hops model with the adjacent weight sharing scheme. For all tasks that output lists (i.e. the answers are multiple words), we take each possible combination of possible outputs and record them as a separate answer vocabulary word.

**Sentence Representation:** In our experiments we explore two different representations for the sentences. The first is the bag-of-words (BoW) representation that takes the sentence $x_i = \{x_{i1}, x_{i2}, ..., x_{in}\}$, embeds each word and sums the resulting vectors: e.g $m_i = \sum_j A x_{ij}$ and $c_i = \sum_j C x_{ij}$. The input vector $u$ representing the question is also embedded as a bag of words: $u = \sum_j B q_j$. This has the drawback that it cannot capture the order of the words in the sentence, which is important for some tasks.

We therefore propose a second representation that encodes the position of words within the sentence. This takes the form: $m_i = \sum_j l_j \cdot A x_{ij}$, where $\cdot$ is an element-wise multiplication. $l_j$ is a

column vector with the structure $l_{kj} = (1 - j/J) - (k/d)(1 - 2j/J)$ (assuming 1-based indexing), with $J$ being the number of words in the sentence, and $d$ is the dimension of the embedding. This sentence representation, which we call position encoding (PE), means that the order of the words now affects $m_i$. The same representation is used for questions, memory inputs and memory outputs.

**Temporal Encoding:** Many of the QA tasks require some notion of temporal context, i.e. in the first example of Section 2, the model needs to understand that Sam is in the bedroom after he is in the kitchen. To enable our model to address them, we modify the memory vector so that $m_i = \sum_j A x_{ij} + T_A(i)$, where $T_A(i)$ is the $i$th row of a special matrix $T_A$ that encodes temporal information. The output embedding is augmented in the same way with a matrix $T_c$ (e.g. $c_i = \sum_j C x_{ij} + T_C(i)$). Both $T_A$ and $T_C$ are learned during training. They are also subject to the same sharing constraints as $A$ and $C$. Note that sentences are indexed in reverse order, reflecting their relative distance from the question so that $x_1$ is the last sentence of the story.

**Learning time invariance by injecting random noise**: we have found it helpful to add "dummy" memories to regularize $T_A$. That is, at training time we can randomly add 10% of empty memories to the stories. We refer to this approach as random noise (RN).

### 4.2 Training Details

10% of the bAbI training set was held-out to form a validation set, which was used to select the optimal model architecture and hyperparameters. Our models were trained using a learning rate of $\eta = 0.01$, with anneals every 25 epochs by $\eta/2$ until 100 epochs were reached. No momentum or weight decay was used. The weights were initialized randomly from a Gaussian distribution with zero mean and $\sigma = 0.1$. When trained on all tasks simultaneously with 1k training samples (10k training samples), 60 epochs (20 epochs) were used with learning rate anneals of $\eta/2$ every 15 epochs (5 epochs). All training uses a batch size of 32 (but cost is not averaged over a batch), and gradients with an $\ell_2$ norm larger than 40 are divided by a scalar to have norm 40. In some of our experiments, we explored commencing training with the softmax in each memory layer removed, making the model entirely linear except for the final softmax for answer prediction. When the validation loss stopped decreasing, the softmax layers were re-inserted and training recommenced. We refer to this as linear start (LS) training. In LS training, the initial learning rate is set to $\eta = 0.005$. The capacity of memory is restricted to the most recent 50 sentences. Since the number of sentences and the number of words per sentence varied between problems, a null symbol was used to pad them all to a fixed size. The embedding of the null symbol was constrained to be zero.

On some tasks, we observed a large variance in the performance of our model (i.e. sometimes failing badly, other times not, depending on the initialization). To remedy this, we repeated each training 10 times with different random initializations, and picked the one with the lowest training error.

### 4.3 Baselines

We compare our approach[2] (abbreviated to MemN2N) to a range of alternate models:

- **MemNN:** The strongly supervised AM+NG+NL Memory Networks approach, proposed in [22]. This is the best reported approach in that paper. It uses a max operation (rather than softmax) at each layer which is trained directly with supporting facts (strong supervision). It employs $n$-gram modeling, nonlinear layers and an adaptive number of hops per query.

- **MemNN-WSH:** A weakly supervised heuristic version of MemNN where the supporting sentence labels are not used in training. Since we are unable to backpropagate through the max operations in each layer, we enforce that the first memory hop should share at least one word with the question, and that the second memory hop should share at least one word with the first hop and at least one word with the answer. All those memories that conform are called valid memories, and the goal during training is to rank them higher than invalid memories using the same ranking criteria as during strongly supervised training.

- **LSTM:** A standard LSTM model, trained using question / answer pairs only (i.e. also weakly supervised). For more detail, see [22].

## 4.4 Results

We report a variety of design choices: (i) BoW vs Position Encoding (PE) sentence representation; (ii) training on all 20 tasks independently vs jointly training (joint training used an embedding dimension of $d = 50$, while independent training used $d = 20$); (iii) two phase training: linear start (LS) where softmaxes are removed initially vs training with softmaxes from the start; (iv) varying memory hops from 1 to 3.

The results across all 20 tasks are given in Table 1 for the 1k training set, along with the mean performance for 10k training set[3]. They show a number of interesting points:

- The best MemN2N models are reasonably close to the supervised models (e.g. 1k: 6.7% for MemNN vs 12.6% for MemN2N with position encoding + linear start + random noise, jointly trained and 10k: 3.2% for MemNN vs 4.2% for MemN2N with position encoding + linear start + random noise + non-linearity[4], although the supervised models are still superior.

- All variants of our proposed model comfortably beat the weakly supervised baseline methods.

- The position encoding (PE) representation improves over bag-of-words (BoW), as demonstrated by clear improvements on tasks 4, 5, 15 and 18, where word ordering is particularly important.

- The linear start (LS) to training seems to help avoid local minima. See task 16 in Table 1, where PE alone gets 53.6% error, while using LS reduces it to 1.6%.

- Jittering the time index with random empty memories (RN) as described in Section 4.1 gives a small but consistent boost in performance, especially for the smaller 1k training set.

- Joint training on all tasks helps.

- Importantly, more computational hops give improved performance. We give examples of the hops performed (via the values of eq. (1)) over some illustrative examples in Fig. 2 and in the supplementary material.

| | Baseline | | | MemN2N | | | | | | | | |
|---|---|---|---|---|---|---|---|---|---|---|---|---|
| Task | Strongly Supervised MemNN [22] | LSTM [22] | MemNN WSH | BoW | PE | PE LS | PE LS RN | 1 hop PE LS joint | 2 hops PE LS joint | 3 hops PE LS joint | PE LS RN joint | PE LS LW joint |
| 1: 1 supporting fact | 0.0 | 50.0 | 0.1 | 0.6 | 0.1 | 0.2 | 0.0 | 0.8 | 0.0 | 0.1 | 0.0 | 0.1 |
| 2: 2 supporting facts | 0.0 | 80.0 | 42.8 | 17.6 | 21.6 | 12.8 | 8.3 | 62.0 | 15.6 | 14.0 | 11.4 | 18.8 |
| 3: 3 supporting facts | 0.0 | 80.0 | 76.4 | 71.0 | 64.2 | 58.8 | 40.3 | 76.9 | 31.6 | 33.1 | 21.9 | 31.7 |
| 4: 2 argument relations | 0.0 | 39.0 | 40.3 | 32.0 | 3.8 | 11.6 | 2.8 | 22.8 | 2.2 | 5.7 | 13.4 | 17.5 |
| 5: 3 argument relations | 2.0 | 30.0 | 16.3 | 18.3 | 14.1 | 15.7 | 13.1 | 11.0 | 13.4 | 14.8 | 14.4 | 12.9 |
| 6: yes/no questions | 0.0 | 52.0 | 51.0 | 8.7 | 7.9 | 8.7 | 7.6 | 7.2 | 2.3 | 3.3 | 2.8 | 2.0 |
| 7: counting | 15.0 | 51.0 | 36.1 | 23.5 | 21.6 | 20.3 | 17.3 | 15.9 | 25.4 | 17.9 | 18.3 | 10.1 |
| 8: lists/sets | 9.0 | 55.0 | 37.8 | 11.4 | 12.6 | 12.7 | 10.0 | 13.2 | 11.7 | 10.1 | 9.3 | 6.1 |
| 9: simple negation | 0.0 | 36.0 | 35.9 | 21.1 | 23.3 | 17.0 | 13.2 | 5.1 | 2.0 | 3.1 | 1.9 | 1.5 |
| 10: indefinite knowledge | 2.0 | 56.0 | 68.7 | 22.8 | 17.4 | 18.6 | 15.1 | 10.6 | 5.0 | 6.6 | 6.5 | 2.6 |
| 11: basic coreference | 0.0 | 38.0 | 30.0 | 4.1 | 4.3 | 0.0 | 0.9 | 8.4 | 1.2 | 0.9 | 0.3 | 3.3 |
| 12: conjunction | 0.0 | 26.0 | 10.1 | 0.3 | 0.3 | 0.1 | 0.2 | 0.4 | 0.0 | 0.3 | 0.1 | 0.0 |
| 13: compound coreference | 0.0 | 6.0 | 19.7 | 10.5 | 9.9 | 0.3 | 0.4 | 6.3 | 0.2 | 1.4 | 0.2 | 0.5 |
| 14: time reasoning | 1.0 | 73.0 | 18.3 | 1.3 | 1.8 | 2.0 | 1.7 | 36.9 | 8.1 | 8.2 | 6.9 | 2.0 |
| 15: basic deduction | 0.0 | 79.0 | 64.8 | 24.3 | 0.0 | 0.0 | 0.0 | 46.4 | 0.5 | 0.0 | 0.0 | 1.8 |
| 16: basic induction | 0.0 | 77.0 | 50.5 | 52.0 | 52.1 | 1.6 | 1.3 | 47.4 | 51.3 | 3.5 | 2.7 | 51.0 |
| 17: positional reasoning | 35.0 | 49.0 | 50.9 | 45.4 | 50.1 | 49.0 | 51.0 | 44.4 | 41.2 | 44.5 | 40.4 | 42.6 |
| 18: size reasoning | 5.0 | 48.0 | 51.3 | 48.1 | 13.6 | 10.1 | 11.1 | 9.6 | 10.3 | 9.2 | 9.4 | 9.2 |
| 19: path finding | 64.0 | 92.0 | 100.0 | 89.7 | 87.4 | 85.6 | 82.8 | 90.7 | 89.9 | 90.2 | 88.0 | 90.6 |
| 20: agent's motivation | 0.0 | 9.0 | 3.6 | 0.1 | 0.0 | 0.0 | 0.0 | 0.0 | 0.1 | 0.0 | 0.0 | 0.2 |
| Mean error (%) | 6.7 | 51.3 | 40.2 | 25.1 | 20.3 | 16.3 | 13.9 | 25.8 | 15.6 | 13.3 | 12.4 | 15.2 |
| Failed tasks (err. > 5%) | 4 | 20 | 18 | 15 | 13 | 12 | 11 | 17 | 11 | 11 | 11 | 10 |
| On 10k training data | | | | | | | | | | | | |
| Mean error (%) | 3.2 | 36.4 | 39.2 | 15.4 | 9.4 | 7.2 | 6.6 | 24.5 | 10.9 | 7.9 | 7.5 | 11.0 |
| Failed tasks (err. > 5%) | 2 | 16 | 17 | 9 | 6 | 4 | 4 | 16 | 7 | 6 | 6 | 6 |

Table 1: Test error rates (%) on the 20 QA tasks for models using 1k training examples (mean test errors for 10k training examples are shown at the bottom). Key: BoW = bag-of-words representation; PE = position encoding representation; LS = linear start training; RN = random injection of time index noise; LW = RNN-style layer-wise weight tying (if not stated, adjacent weight tying is used); joint = joint training on all tasks (as opposed to per-task training).

## 5 Language Modeling Experiments

The goal in language modeling is to predict the next word in a text sequence given the previous words $x$. We now explain how our model can easily be applied to this task.

| Story (1: 1 supporting fact) | Support | Hop 1 | Hop 2 | Hop 3 |
|---|---|---|---|---|
| Daniel went to the bathroom. | | 0.00 | 0.00 | 0.03 |
| Mary travelled to the hallway. | | 0.00 | 0.00 | 0.00 |
| John went to the bedroom. | | 0.37 | 0.02 | 0.00 |
| John travelled to the bathroom. | yes | 0.60 | 0.98 | 0.96 |
| Mary went to the office. | | 0.01 | 0.00 | 0.00 |
| **Where is John?  Answer: bathroom   Prediction: bathroom** | | | | |

| Story (2: 2 supporting facts) | Support | Hop 1 | Hop 2 | Hop 3 |
|---|---|---|---|---|
| John dropped the milk. | | 0.06 | 0.00 | 0.00 |
| John took the milk there. | yes | 0.88 | 1.00 | 0.00 |
| Sandra went back to the bathroom. | | 0.00 | 0.00 | 0.00 |
| John moved to the hallway. | yes | 0.00 | 0.00 | 1.00 |
| Mary went back to the bedroom. | | 0.00 | 0.00 | 0.00 |
| **Where is the milk?  Answer: hallway   Prediction: hallway** | | | | |

| Story (16: basic induction) | Support | Hop 1 | Hop 2 | Hop 3 |
|---|---|---|---|---|
| Brian is a frog. | yes | 0.00 | 0.98 | 0.00 |
| Lily is gray. | | 0.07 | 0.00 | 0.00 |
| Brian is yellow. | yes | 0.07 | 0.00 | 1.00 |
| Julius is green. | | 0.06 | 0.00 | 0.00 |
| Greg is a frog. | yes | 0.76 | 0.02 | 0.00 |
| **What color is Greg?  Answer: yellow   Prediction: yellow** | | | | |

| Story (18: size reasoning) | Support | Hop 1 | Hop 2 | Hop 3 |
|---|---|---|---|---|
| The suitcase is bigger than the chest. | yes | 0.00 | 0.88 | 0.00 |
| The box is bigger than the chocolate. | | 0.04 | 0.05 | 0.10 |
| The chest is bigger than the chocolate. | yes | 0.17 | 0.07 | 0.90 |
| The chest fits inside the container. | | 0.00 | 0.00 | 0.00 |
| The chest fits inside the box. | | 0.00 | 0.00 | 0.00 |
| **Does the suitcase fit in the chocolate?  Answer: no   Prediction: no** | | | | |

Figure 2: Example predictions on the QA tasks of [22]. We show the labeled supporting facts (support) from the dataset which MemN2N does not use during training, and the probabilities $p$ of each hop used by the model during inference. MemN2N successfully learns to focus on the correct supporting sentences.

| | Penn Treebank | | | | | Text8 | | | | |
|---|---|---|---|---|---|---|---|---|---|---|
| Model | # of hidden | # of hops | memory size | Valid. perp. | Test perp. | # of hidden | # of hops | memory size | Valid. perp. | Test perp. |
| RNN [15] | 300 | - | - | 133 | 129 | 500 | - | - | - | 184 |
| LSTM [15] | 100 | - | - | 120 | 115 | 500 | - | - | 122 | 154 |
| SCRN [15] | 100 | - | - | 120 | 115 | 500 | - | - | - | 161 |
| MemN2N | 150 | 2 | 100 | 128 | 121 | 500 | 2 | 100 | 152 | 187 |
| | 150 | 3 | 100 | 129 | 122 | 500 | 3 | 100 | 142 | 178 |
| | 150 | 4 | 100 | 127 | 120 | 500 | 4 | 100 | 129 | 162 |
| | 150 | 5 | 100 | 127 | 118 | 500 | 5 | 100 | 123 | 154 |
| | 150 | 6 | 100 | 122 | 115 | 500 | 6 | 100 | 124 | 155 |
| | 150 | 7 | 100 | 120 | 114 | 500 | 7 | 100 | 118 | **147** |
| | 150 | 6 | 25 | 125 | 118 | 500 | 6 | 25 | 131 | 163 |
| | 150 | 6 | 50 | 121 | 114 | 500 | 6 | 50 | 132 | 166 |
| | 150 | 6 | 75 | 122 | 114 | 500 | 6 | 75 | 126 | 158 |
| | 150 | 6 | 100 | 122 | 115 | 500 | 6 | 100 | 124 | 155 |
| | 150 | 6 | 125 | 120 | 112 | 500 | 6 | 125 | 125 | 157 |
| | 150 | 6 | 150 | 121 | 114 | 500 | 6 | 150 | 123 | 154 |
| | 150 | 7 | 200 | 118 | **111** | - | - | - | - | - |

Table 2: The perplexity on the test sets of Penn Treebank and Text8 corpora. Note that increasing the number of memory hops improves performance.

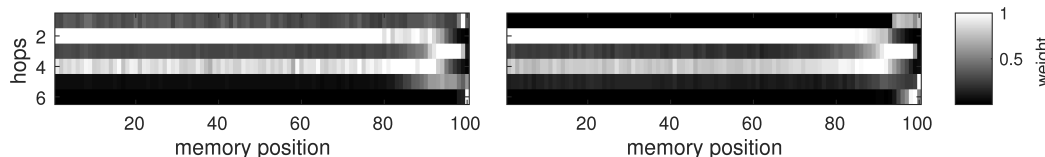

Figure 3: Average activation weight of memory positions during 6 memory hops. White color indicates where the model is attending during the $k^{th}$ hop. For clarity, each row is normalized to have maximum value of 1. A model is trained on (left) Penn Treebank and (right) Text8 dataset.

We now operate on word level, as opposed to the sentence level. Thus the previous $N$ words in the sequence (including the current) are embedded into memory separately. Each memory cell holds only a single word, so there is no need for the BoW or linear mapping representations used in the QA tasks. We employ the temporal embedding approach of Section 4.1.

Since there is no longer any question, $q$ in Fig. 1 is fixed to a constant vector 0.1 (without embedding). The output softmax predicts which word in the vocabulary (of size $V$) is next in the sequence. A cross-entropy loss is used to train model by backpropagating the error through multiple memory layers, in the same manner as the QA tasks. To aid training, we apply ReLU operations to half of the units in each layer. We use layer-wise (RNN-like) weight sharing, i.e. the query weights of each layer are the same; the output weights of each layer are the same. As noted in Section 2.2, this makes our architecture closely related to an RNN which is traditionally used for language

modeling tasks; however here the "sequence" over which the network is recurrent is not in the text, but in the memory hops. Furthermore, the weight tying restricts the number of parameters in the model, helping generalization for the deeper models which we find to be effective for this task. We use two different datasets:

**Penn Tree Bank** [13]: This consists of 929k/73k/82k train/validation/test words, distributed over a vocabulary of 10k words. The same preprocessing as [25] was used.

**Text8** [15]: This is a a pre-processed version of the first 100M million characters, dumped from Wikipedia. This is split into 93.3M/5.7M/1M character train/validation/test sets. All word occurring less than 5 times are replaced with the <UNK> token, resulting in a vocabulary size of ∼44k.

## 5.1 Training Details

The training procedure we use is the same as the QA tasks, except for the following. For each mini-batch update, the $\ell_2$ norm of the whole gradient of all parameters is measured[5] and if larger than $L = 50$, then it is scaled down to have norm $L$. This was crucial for good performance. We use the learning rate annealing schedule from [15], namely, if the validation cost has not decreased after one epoch, then the learning rate is scaled down by a factor 1.5. Training terminates when the learning rate drops below $10^{-5}$, i.e. after 50 epochs or so. Weights are initialized using $\mathcal{N}(0, 0.05)$ and batch size is set to 128. On the Penn tree dataset, we repeat each training 10 times with different random initializations and pick the one with smallest validation cost. However, we have done only a single training run on Text8 dataset due to limited time constraints.

## 5.2 Results

Table 2 compares our model to RNN, LSTM and Structurally Constrained Recurrent Nets (SCRN) [15] baselines on the two benchmark datasets. Note that the baseline architectures were tuned in [15] to give optimal perplexity[6]. Our MemN2N approach achieves lower perplexity on both datasets (111 vs 115 for RNN/SCRN on Penn and 147 vs 154 for LSTM on Text8). Note that MemN2N has ∼1.5x more parameters than RNNs with the same number of hidden units, while LSTM has ∼4x more parameters. We also vary the number of hops and memory size of our MemN2N, showing the contribution of both to performance; note in particular that increasing the number of hops helps. In Fig. 3, we show how MemN2N operates on memory with multiple hops. It shows the average weight of the activation of each memory position over the test set. We can see that some hops concentrate only on recent words, while other hops have more broad attention over all memory locations, which is consistent with the idea that succeful language models consist of a smoothed $n$-gram model and a cache [15]. Interestingly, it seems that those two types of hops tend to alternate. Also note that unlike a traditional RNN, the cache does not decay exponentially: it has roughly the same average activation across the entire memory. This may be the source of the observed improvement in language modeling.

# 6 Conclusions and Future Work

In this work we showed that a neural network with an explicit memory and a recurrent attention mechanism for reading the memory can be successfully trained via backpropagation on diverse tasks from question answering to language modeling. Compared to the Memory Network implementation of [23] there is no supervision of supporting facts and so our model can be used in a wider range of settings. Our model approaches the same performance of that model, and is significantly better than other baselines with the same level of supervision. On language modeling tasks, it slightly outperforms tuned RNNs and LSTMs of comparable complexity. On both tasks we can see that increasing the number of memory hops improves performance.

However, there is still much to do. Our model is still unable to exactly match the performance of the memory networks trained with strong supervision, and both fail on several of the 1k QA tasks. Furthermore, smooth lookups may not scale well to the case where a larger memory is required. For these settings, we plan to explore multiscale notions of attention or hashing, as proposed in [23].

## Acknowledgments

The authors would like to thank Armand Joulin, Tomas Mikolov, Antoine Bordes and Sumit Chopra for useful comments and valuable discussions, and also the FAIR Infrastructure team for their help and support.

## Footnotes

[1]Note that in this view, the terminology of input and output from Fig. 1 is flipped - when viewed as a traditional RNN with this special conditioning of outputs, $A$ becomes part of the output embedding of the RNN and $C$ becomes the input embedding.

[2] MemN2N source code is available at `https://github.com/facebook/MemNN`.

[3]More detailed results for the 10k training set can be found in the supplementary material.

[4]Following [17] we found adding more non-linearity solves tasks 17 and 19, see the supplementary material.

[5]In the QA tasks, the gradient of each weight matrix is measured separately.

[6]They tuned the hyper-parameters on Penn Treebank and used them on Text8 without additional tuning, except for the number of hidden units. See [15] for more detail.

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
