[Supplementary Material]

# End-To-End Memory Networks Supplementary Material

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

Where is the milk? Answer: hallway Prediction: hallway

**Story (3: 3 supporting facts)**

| | Support | Hop 1 | Hop 2 | Hop 3 |
|---|---|---|---|---|
| John moved to the hallway. | | 0.00 | 0.00 | 0.00 |
| John grabbed the football. | yes | 0.00 | 1.00 | 0.00 |
| John journeyed to the garden. | | 0.35 | 0.00 | 0.00 |
| Sandra moved to the hallway. | | 0.00 | 0.00 | 0.00 |
| John went back to the hallway. | yes | 0.00 | 0.00 | 1.00 |
| John journeyed to the garden. | yes | 0.62 | 0.00 | 0.00 |

Where was the football before the garden? A: hallway P: hallway

**Story (4: 2 argument relations)**

| | Support | Hop 1 | Hop 2 | Hop 3 |
|---|---|---|---|---|
| The garden is north of the kitchen. | yes | 0.84 | 1.00 | 0.92 |
| The kitchen is north of the bedroom. | | 0.16 | 0.00 | 0.08 |

What is north of the kitchen? Answer: garden Prediction: garden

**Story (5: 3 argument relations)**

| | Support | Hop 1 | Hop 2 | Hop 3 |
|---|---|---|---|---|
| Jeff travelled to the bedroom. | | 0.00 | 0.00 | 0.00 |
| Jeff journeyed to the garden. | | 0.00 | 0.00 | 0.00 |
| Fred handed the apple to Jeff. | yes | 1.00 | 1.00 | 0.98 |
| Mary went to the garden. | | 0.00 | 0.00 | 0.00 |
| Fred went back to the bathroom. | | 0.00 | 0.00 | 0.00 |
| Fred got the milk there. | | 0.00 | 0.00 | 0.00 |
| Mary journeyed to the kitchen. | | 0.00 | 0.00 | 0.00 |

Who gave the apple to Jeff? Answer: Fred Prediction: Fred

**Story (6: yes/no questions)**

| | Support | Hop 1 | Hop 2 | Hop 3 |
|---|---|---|---|---|
| Sandra travelled to the bedroom. | | 0.06 | 0.00 | 0.01 |
| John took the football there. | | 0.00 | 0.00 | 0.00 |
| Sandra travelled to the office. | | 0.00 | 0.45 | 0.16 |
| Sandra went to the bedroom. | yes | 0.89 | 0.39 | 0.04 |
| Daniel went back to the kitchen. | | 0.00 | 0.16 | 0.00 |
| John took the apple there. | | 0.00 | 0.00 | 0.00 |
| Mary got the milk there. | | 0.00 | 0.00 | 0.00 |

Is Sandra in the bedroom? Answer: yes Prediction: Yes

**Story (7: counting)**

| | Support | Hop 1 | Hop 2 | Hop 3 |
|---|---|---|---|---|
| Daniel moved to the office. | | 0.00 | 0.00 | 0.00 |
| Mary moved to the office. | | 0.00 | 0.00 | 0.00 |
| Sandra picked up the apple there. | yes | 0.14 | 0.00 | 0.92 |
| Sandra dropped the apple. | yes | 0.12 | 0.00 | 0.00 |
| Sandra took the apple there. | yes | 0.73 | 1.00 | 0.08 |
| John went to the bedroom. | | 0.00 | 0.00 | 0.00 |

How many objects is Sandra carrying? Answer: one Prediction: one

**Story (8: lists/sets)**

| | Support | Hop 1 | Hop 2 | Hop 3 |
|---|---|---|---|---|
| John moved to the hallway. | | 0.00 | 0.00 | 0.00 |
| John journeyed to the garden. | | 0.00 | 0.00 | 0.00 |
| Daniel moved to the garden. | | 0.00 | 0.01 | 0.00 |
| Daniel grabbed the apple there. | yes | 0.03 | 0.00 | 0.98 |
| Daniel got the milk there. | yes | 0.97 | 0.02 | 0.00 |
| John went back to the hallway. | | 0.00 | 0.00 | 0.00 |

What is Daniel carrying? Answer: apple,milk Prediction: apple,milk

**Story (9: simple negation)**

| | Support | Hop 1 | Hop 2 | Hop 3 |
|---|---|---|---|---|
| Sandra is in the garden. | | 0.60 | 0.99 | 0.00 |
| Sandra is not in the garden. | yes | 0.37 | 0.01 | 1.00 |
| John went to the office. | | 0.00 | 0.00 | 0.00 |
| John is in the bedroom. | | 0.00 | 0.00 | 0.00 |
| Daniel moved to the garden. | | 0.00 | 0.00 | 0.00 |

Is Sandra in the garden? Answer: no Prediction: no

**Story (10: indefinite knowledge)**

| | Support | Hop 1 | Hop 2 | Hop 3 |
|---|---|---|---|---|
| Julie is either in the school or the bedroom. | | 0.00 | 0.00 | 0.00 |
| Julie is either in the cinema or the park. | | 0.00 | 0.00 | 0.00 |
| Bill is in the park. | | 0.00 | 0.00 | 0.00 |
| Bill is either in the office or the office. | yes | 1.00 | 1.00 | 1.00 |

Is Bill in the office? Answer: maybe Prediction: maybe

**Story (11: basic coherence)**

| | Support | Hop 1 | Hop 2 | Hop 3 |
|---|---|---|---|---|
| Mary journeyed to the hallway. | | 0.00 | 0.01 | 0.00 |
| After that she journeyed to the bathroom. | | 0.00 | 0.00 | 0.00 |
| Mary journeyed to the garden. | | 0.00 | 0.00 | 0.00 |
| Then she went to the office. | | 0.01 | 0.06 | 0.00 |
| Sandra journeyed to the garden. | yes | 0.97 | 0.42 | 0.00 |
| Then she went to the hallway. | yes | 0.00 | 0.50 | 1.00 |

Where is Sandra? Answer: hallway Prediction: hallway

**Story (12: conjunction)**

| | Support | Hop 1 | Hop 2 | Hop 3 |
|---|---|---|---|---|
| John and Sandra went back to the kitchen. | | 0.08 | 0.00 | 0.00 |
| Sandra and Mary travelled to the garden. | | 0.05 | 0.00 | 0.00 |
| Mary and Daniel travelled to the office. | | 0.00 | 0.00 | 0.00 |
| Mary and John went to the bathroom. | | 0.01 | 0.00 | 0.00 |
| Daniel and Sandra went to the kitchen. | yes | 0.74 | 1.00 | 1.00 |
| Daniel and Mary journeyed to the office. | | 0.06 | 0.00 | 0.00 |

Where is Sandra? Answer: kitchen Prediction: kitchen

**Story (13: compound coherence)**

| | Support | Hop 1 | Hop 2 | Hop 3 |
|---|---|---|---|---|
| Sandra and Daniel travelled to the bathroom. | | 0.13 | 0.00 | 0.00 |
| Afterwards they went back to the office. | | 0.01 | 0.00 | 0.00 |
| Daniel and Mary travelled to the hallway. | | 0.01 | 0.00 | 0.00 |
| Following that they went back to the office. | | 0.06 | 0.04 | 0.00 |
| Mary and Sandra moved to the hallway. | yes | 0.59 | 0.02 | 0.00 |
| Then they went to the kitchen. | yes | 0.02 | 0.94 | 1.00 |

Where is Sandra? Answer: kitchen Prediction: kitchen

**Story (14: time reasoning)**

| | Support | Hop 1 | Hop 2 | Hop 3 |
|---|---|---|---|---|
| This morning Julie went to the cinema. | | 0.00 | 0.03 | 0.00 |
| Julie journeyed to the kitchen yesterday. | | 0.00 | 0.04 | 0.01 |
| Fred travelled to the cinema yesterday. | | 0.00 | 0.05 | 0.01 |
| Bill travelled to the office yesterday. | | 0.00 | 0.07 | 0.01 |
| This morning Mary travelled to the bedroom. | yes | 0.97 | 0.27 | 0.01 |
| Yesterday Mary journeyed to the cinema. | yes | 0.01 | 0.33 | 0.96 |

Where was Mary before the bedroom? Answer: cinema Prediction: cinema

**Story (15: basic deduction)**

| | Support | Hop 1 | Hop 2 | Hop 3 |
|---|---|---|---|---|
| Cats are afraid of wolves. | yes | 0.00 | 0.99 | 0.62 |
| Sheep are afraid of wolves. | | 0.00 | 0.00 | 0.31 |
| Winona is a sheep. | | 0.00 | 0.00 | 0.00 |
| Emily is a sheep. | | 0.00 | 0.00 | 0.00 |
| Gertrude is a cat. | yes | 0.99 | 0.00 | 0.00 |
| Wolves are afraid of mice. | | 0.00 | 0.00 | 0.00 |
| Mice are afraid of wolves. | | 0.00 | 0.00 | 0.07 |
| Jessica is a mouse. | | 0.00 | 0.00 | 0.00 |

What is gertrude afraid of? Answer: wolf Prediction: wolf

**Story (16: basic induction)**

| | Support | Hop 1 | Hop 2 | Hop 3 |
|---|---|---|---|---|
| Lily is a swan. | | 0.00 | 0.00 | 0.00 |
| Brian is a frog. | yes | 0.00 | 0.98 | 0.00 |
| Lily is gray. | | 0.07 | 0.00 | 0.00 |
| Brian is yellow. | yes | 0.07 | 0.00 | 1.00 |
| Julius is a swan. | | 0.00 | 0.00 | 0.00 |
| Bernhard is yellow. | | 0.04 | 0.00 | 0.00 |
| Julius is green. | | 0.06 | 0.00 | 0.00 |
| Greg is a frog. | yes | 0.76 | 0.02 | 0.00 |

What color is Greg? Answer: yellow Prediction: yellow

**Story (17: positional reasoning)**

| | Support | Hop 1 | Hop 2 | Hop 3 |
|---|---|---|---|---|
| The red square is below the red sphere. | yes | 0.37 | 0.95 | 0.58 |
| The red sphere is below the triangle. | yes | 0.63 | 0.05 | 0.43 |

Is the triangle above the red square? Answer: yes Prediction: no

**Story (18: size reasoning)**

| | Support | Hop 1 | Hop 2 | Hop 3 |
|---|---|---|---|---|
| The suitcase is bigger than the chest. | yes | 0.00 | 0.88 | 0.00 |
| The box is bigger than the chocolate. | | 0.04 | 0.05 | 0.10 |
| The chest is bigger than the chocolate. | yes | 0.17 | 0.07 | 0.90 |
| The chest fits inside the container. | | 0.00 | 0.00 | 0.00 |
| The chest fits inside the box. | | 0.00 | 0.00 | 0.00 |

Does the suitcase fit in the chocolate? Answer: no Prediction: no

**Story (19: path finding)**

| | Support | Hop 1 | Hop 2 | Hop 3 |
|---|---|---|---|---|
| The hallway is north of the kitchen. | | 1.00 | 1.00 | 1.00 |
| The garden is south of the kitchen. | yes | 0.00 | 0.00 | 0.00 |
| The garden is east of the bedroom. | yes | 0.00 | 0.00 | 0.00 |
| The bathroom is south of the bedroom. | | 0.00 | 0.00 | 0.00 |
| The office is east of the garden. | | 0.00 | 0.00 | 0.00 |

How do you go from the kitchen to the bedroom? Answer: s,w Prediction: n,n

**Story (20: agent's motivation)**

| | Support | Hop 1 | Hop 2 | Hop 3 |
|---|---|---|---|---|
| Yann journeyed to the kitchen. | | 0.00 | 0.00 | 0.00 |
| Yann grabbed the apple there. | | 0.00 | 0.00 | 0.00 |
| Antoine is thirsty. | yes | 0.17 | 0.00 | 0.98 |
| Jason picked up the milk there. | | 0.01 | 0.00 | 0.00 |
| Antoine travelled to the kitchen. | | 0.77 | 1.00 | 0.00 |

Why did antoine go to the kitchen? Answer: thirsty Prediction: thirsty

Figure 1: Examples of attention weights during different memory hops for the bAbi tasks. The model is PE+LS+RN with 3 memory hops that is trained separately on each task with 10k training data. The support column shows which sentences are necessary for answering questions. Although this information is not used, the model succesfully learns to focus on the correct support sentences on most of the tasks. The hop columns show where the model put more weight (indicated by values and blue color) during its three hops. The mistakes made by the model are highlighted by red color.