[Reviews · NeurIPS 2015]

Submitted by Assigned_Reviewer_1

The authors propose new end-to-end memory networks which can be applied to QA or language modeling. The proposed networks using attention model on external memory are compared with the previous attention based models. In QA and language model, the proposed networks provide better accuracy and prediction abilities.

Quality:

- The manuscript addresses an important issue, attention on external memory. However, the model needs some rationale on which the model has been designed.

Clarity:

- Some arguments are not clear. For example, the proposed model is compared with standard RNN with internal and external outputs. However this comparison seems not clear. Basically section 3 could've been clearer with some equations.

- The objective function for the linear start training needs to be clarified.

- It would be better if Figures 2 and 3 have clearer explanations.

Originality:

- The proposed model is original but the contribution seems not significant, since there have been many attention mechanisms on external memory.

Significance:

- The topic is quite significant but the results are not strong enough to be persuasive. The experiments were conducted with small data sets and the results are not consistent as the authors mentioned. Also, with the language model, the perp. of SCRN (or LSTM) with 100 nodes seems very similar to the one of MemN2N with 150 nodes.

Summary: The authors proposed new end-to-end memory networks. However, the proposed networks seem not well designed.

Submitted by Assigned_Reviewer_2

Summary:

This paper presents an end-to-end version of memory networks (Weston et al., 2015) such that the model doesn't train on the intermediate 'supporting facts' strong supervision of which input sentences are the best memory accesses, making it much more realistic. They also have multiple hops (computational steps) per output symbol. The tasks are Q&A and language modeling, and achieves strong results.

Quality: The paper is a useful extension of memNN because it removes the strong, unrealistic supervision requirement and still performs pretty competitively. The architecture is defined pretty cleanly and simply. The related work section is quite well-written, detailing the various similarities and differences with multiple streams of related work. The discussion about the model's connection to RNNs is also useful.

My main concern with the paper is that it presents various model decisions and choices without motivating them, without empirically ablating them, and also without apparently using a dev set for most of these decisions (and hence tuning and presenting too many results on the test set directly?). Some examples are: -- Eqn3 -- the various wight tying choices, and whether these generalize to other tasks (Q&A or otherwise) -- temporal encoding design choices -- the various training decisions in Sec 4.2 and Sec 4.4 -- Table 1, that does finally have ablations for some remaining, more important choices, is done directly on the test set, at the risk of reporting too many numbers on the test set and essentially choosing the best among those as their final number to compare to baselines. Ideally, all these decisions should be made on a dev/validation set and then the final setting choice should be run on test.

Other issues or suggestions: -- Encoding with an RNN or LSTM-RNN would be good to compare to the current approach of BoW or temporal encoding. -- Also, for the MemNN-WSH baseline, why not have a better classifier to predict which sentences to use, instead of using >1 word match? -- the perplexity differences in Table2 from the LSTM baseline do not look large so authors should report statistical significance results. The gap might be even smaller or negative if the current model did not so many extra parameters to tune over, as compared to LSTMs. -- perplexity in language modeling is not very convincing as a metric, so maybe useful to also apply this model (and show its generalizability) to text categorization tasks like tagging, NER, sentiment, etc.

Clarity: The paper is well-written and well-organized

Originality and significance: The paper is original and has a useful impact in terms of moving away from the strong supervision assumption of memNNs, and using multiple hops per output; the related work is well-written to clarify the originality. However, there are issues with model tuning and decisions, comparison, and need of better tasks.

########## Post-rebuttal: Thanks for answering some of my questions. I have increased my score to a 7 but please include all the dev/test details, model choice ablations, and statistical significance tests for perplexity. Also try to add some non-perplexity metrics.
Summary: The paper presents an end-to-end version of memory networks and moves away from the strong supervision assumption of memNNs, making it much more realistic. Using multiple hops per output is also a useful contribution; the related work and connection to RNNs is useful. Empirically, they achieve competitive results as compared to strongly-supervised memNNs, but are almost same as LSTMs on language modeling. Importantly, there are issues with model tuning and decisions (too many choices made without motivation, without empirical ablation, and also without apparently using a dev set for most of these decisions), better baselines, and need of better tasks (instead of or in addition to language modeling and to show better generalization).

Submitted by Assigned_Reviewer_3

This paper propose a new variant of memory networks which does not need supervision during training. One of the strongest points of the paper is the very good job of comparing the proposed model with other prior works, how it differs from them and how it could be interpreted as one of them. The proposed model can not outperform the supervised memory network in the Q&A task but outperform the semi-supervised version, and do very well in language modelling task. The recent trend in having an external memory to capture more long-term dependencies and reasoning is one of the most interesting new avenues in deep learning and this paper is a promising and new approach in this field.

Some detail comments: 50: different notation style for a, q, a would help as i.e. a for answer is indistinguishable from a as an article. 51: The memory space that store unmapped x_i and the one that stores

m_i (L 59) both are referred to as memory. Perhaps it would help readers, if different names for these two memory spaces is used. Fig 1: One thing that is a bit confusing in Fig 1 is the empty slots in the mapped memory locations (coloured blue and beige). Form what understood from the text memory size would be same as the size of input set {x_1, .. x_i}. If that is the case I do not understand the empty white spaces in between. Or are they correspond to them random noises injected as described in L223? -Perhaps this is more future work, but does author have any proposal for the scenario that number of supporting sentences is larger than memory locations? Any mechanism for deleting memory slots? - Did authors try any non-linearity in any part of the model (expect the softmax)? 221: An example with numerical values could help the reader better understand l_{kj} 225: How T_A and T_C are initialized? Did the authors find that the any particular initialization scheme is important for any of the weight matrices? Or just only tired the gaussian initialization as explained in L240? -Did the authors tried to embed the sentences using RNN instead of BoW or PE? 375: half of the units? could you please elaborate more about the reasoning behind this?
Summary: A new variant of memory networks in which it does not need supervision unlike its precedent. A well written and very interesting paper to read with promising results.

Submitted by Assigned_Reviewer_4

Can you give an intuition for the equation in line 221?

I find the temporal encoding and the weighted word vector averaging not very elegant. You are essentially hardcoding neural features at this point.

I think a much more elegant solution would use an RNN for this.

I'm not sure that just because you drop supporting fact supervision your "model can be used in more realistic QA settings". The dataset is still not "realistic" and I would not make such a claim in a paper that only uses babI.
Summary: This paper extends memory networks by adding a multi-hop RNN on top of softmax weighted input (e.g. sentence) vectors.

While experimental results on the synthetic babI dataset are not nearly as good as with the supervised fact setting and language modeling results only show small improvements over (potentially not very well tuned) LSTM baselines, I think the model is very interesting and this is of interest to researchers in deep learning right now.

Author Feedback
Author rebuttal: We thank all the reviewers for their constructive comments.

Assigned_Reviewer_1 (Score:6, Conf:3):
- "The proposed model is original but the contribution seems not significant, since there have been many attention mechanisms on external memory": The previous work on memory-based attention only uses a *single* hop, in contrast to our novel multi-hop model. This non-trivial contribution affords our model far greater power: our experiments show it consistently outperforming single hop approaches.
- "results are not strong enough...experiments were conducted with small data sets": We respectfully disagree, having evaluated our model on the 100M character Text8 dataset, as well as the widely used Penn Tree Bank.
- "the perp. of SCRN (or LSTM) with 100 nodes seems similar to...MemN2N with 150 nodes": As mentioned in the paper (line 405), the LSTM has at least twice more parameters than an MemN2N for the same number of hidden units. Increasing the size of the LSTM does not help either: with 300 hidden units, it overfits to the data and gives worse test perplexity (Mikolov et.al, 2015).
- "results are not consistent as the authors mentioned": Although we observed variance in the model's performance, the validation and test performance were always consistent. Thus good test performance can reliably be achieved by selecting the model with the best validation performance.
- "Basically section 3 could've been clearer": Will revise in final version.
- "The objective function for the linear start training needs to be clarified": During a linear start, the same cross entropy loss is optimized. The only change was that softmax layers in the memory hops are replaced with an identity function.

AR5 (S:5,C:4):
- "the model is still too simple and rigid to take reasoning tasks with reasonable complexity": We respectfully disagree, since experiments show our model is effective at real language modeling tasks, achieving results at least as good as state-of-the-art models of comparable complexity, such as single-layer LSTMs. Certainly, the bAbI tasks are simpler, but existing models such as LSTMs perform poorly on them, unlike our model.

AR4 (S:6,C:4):
- "The gap might be even smaller or negative if the current model did not so many extra parameters to tune over, as compared to LSTMs": LSTM also has many hyper-parameters (e.g. # of steps to unroll, frequency of back-propagation, bias for forget gate etc.). Furthermore, we carefully optimized these to ensure a fair comparison to our model.
- "... without apparently using a dev set for most of these decisions": all design choices were made using a validation set -- none involved test data. For bAbI, 10% of training data was set aside for this. For Text8 and PennTB, the standard val. set was used. We will make this clearer in the final version.

AR3 (S:7,C:5):
- "My main concern with the paper is that it presents various model decisions... without apparently using a dev set for most of these decisions (and hence tuning and presenting too many results on the test set directly?)", "Table 1 ... is done directly on the test set": This is not correct: all design choices were made using a validation set -- none involved test data. For bAbI, 10% of training data was set aside for this. For Text8 and PennTB, the standard val. set was used. We will make this clearer in the final version.
- "presents various model decisions and choices without motivating them, without empirically ablating them": Tables 1 and 2 explore the effects of important design choices, with many variants being shown. The choice of the final model was made using the validation set.
- "the various training decisions in Sec 4.2 and Sec 4.4": We note that many deep learning approaches involve similar training choices and like these, our selections were made to optimize validation performance.
- "intuition for l.221?": The goal is to encode word position, so that we can distinguish "John saw David" from "David saw John". We will clarify this in the final version.
- "...use an RNN for this": Yes, this is a good idea. Unfortunately, the bAbI data was too small to train an RNN, but we are currently exploring this for the language modeling.
- "would not make such a claim in a paper that only uses babI": We will tone down the claim.

AR2 (S:9,C:5) & AR6 (S:7,C:4):
- We appreciate the positive comments and will revise the final version to incorporate the clarity points raised.

AR2:
- "One thing that is a bit confusing in Fig 1 is the empty slots...": The colored memory slots in Fig 1 represent memory locations that are being attended to by the model. So white spaces represent memories that are not attended, not empty memories.
- "Did authors try any non-linearity in any part of the model (expect the softmax)?": Yes. In the language modeling experiments, we applied ReLU nonlinearity to the model state u after each hop, which improved performance.